# Misalignment Effects on Power Gathered by Optical Fiber Pyrometer

**DOI:** 10.3390/s25227011

**Published:** 2025-11-17

**Authors:** Salvador Vargas, Alberto Tapetado, Carmen Vázquez

**Affiliations:** 1Electrical Engineering Faculty, Universidad Tecnológica de Panamá, Ave. Universidad Tecnológica, El Dorado, Panamá 0819-07289, Panama; salvador.vargas@utp.ac.pa; 2Electronics Technology Department, Universidad Carlos III de Madrid, 28911 Leganés, Spain; atapetad@ing.uc3m.es

**Keywords:** metrology, modeling, optical fiber sensor, pyrometer, temperature, lateral displacement, tilting angle, finite hot spot, emissivity, misalignment effects

## Abstract

**Highlights:**

**What are the main findings?**

**What are the implication of the main finding?**

**Abstract:**

This article presents a model for analyzing the effect of misalignments in the optical power gathered by a single-color fiber-optic off-axis pyrometer when there is a tilting angle between the longitudinal fiber axis and the plane of the emitted surface. The model takes into account the fiber parameters, such as the diameter and the numerical aperture, as well as the target object size and measuring distance. Simulations show the simultaneous influence of tilting angle and lateral displacement. This provides key behavioral insights to guide alignment and calibration, helping to avoid temperature measurement errors in advanced manufacturing. The model allows integration of the emissivity angle dependence. Results show that the influence of lateral misalignment is negligible for values below 15 µm (which is 1/10 of the 150 µm minimum distance) when considering a 160 µm object size and a 60° tilting angle. The displacement influence, for a fixed tilting angle, is higher in specific directions of displacement.

## 1. Introduction

Accurate temperature measurements are essential in different industrial applications, from monitoring the temperature in advanced manufacturing, such as grinding [1] and additive manufacturing [2], to foundries and steel production, to ensure high product quality and optimize process efficiency. Thermocouples can measure high temperatures and are welded or embedded on/in the tested sample. However, thermocouple measurements often fail to accurately represent the surface temperature, as thermal equilibrium is difficult to achieve between the thermocouple and the tested sample when the surface temperature is above 400 °C [3]. Being an invasive technique, they are destroyed when used in high-speed machining, where the workpiece is moving at a high speed [4]. Another option is to use non-contact measurement techniques such as Raman thermometry, infrared thermography with cameras, or pyrometry. Raman thermography requires a laser to excite the sample, whose spot delimits the spatial resolution, and the exposure times are usually long [5]. Infrared (IR) thermography with cameras and optical pyrometry requires the use of lenses and the existence of a viewing angle that allows the measurement. Therefore, they can be placed either perpendicular to the measuring surface or at a certain angle, as when using the Wedge method to avoid emissivity dependence of temperature measurements in manufacturing steel strips [6]. An angle is also required when using an off-axis pyrometer in additive manufacturing, where on-axis viewing is blocked or impractical [7,8]. In some applications, being off-axis can also minimize interference from the source (such as the laser in the Laser Powder Bed Fusion process) or other components in the build chamber, leading to more reliable data. In any of those cases, it is very important to ensure a correct alignment of the pyrometer to avoid errors, as the viewed surface and emissivity depend on the viewing angle.

Accuracy validation of surface temperature by optical pyrometry is always a hard task, due, among others, to the uncertainty in knowing the value of the emissivity of the hot object [9]. The analysis of the angular dependence of the surface emissivity in temperature measurements with optical pyrometers shows that for both metals with low emissivity [10] and dielectric materials [11] the influence is low for angles smaller than 40°. Other alignment errors, such as lateral displacement and object size, have not yet been analyzed in detail.

IR thermography with cameras and optical pyrometers requires a viewing angle from the hot spot to the lens that collects the radiation. This lens also determines the spatial resolution of the measurement. The use of optical fiber to directly collect the radiated energy allows the combination of optical fibers of different core and cladding diameters to configure a specific spatial resolution depending on the measuring distance [12]. In any case, a proper alignment of the optical fiber pyrometer to avoid errors in temperature measurements is also required. A model describing the influence of object size on temperature measurements at a specific distance is reported in [13] and experimentally tested in [14]. The influence of placing the fiber-optic pyrometer at a specific angle is analyzed in [15] but still no simultaneous lateral misalignment errors are considered. Fiber optic pyrometry is used in a multitude of applications where those models can be applied, such as machining processes [4], friction monitoring in seismic processes [16], porosity in additive manufacturing [17] and cutting in dry and wet conditions [18] or with electrical discharges [19], among others.

The effect of different misalignments on light coupling between optical fibers, such as angular misalignment alone [20], angular misalignment but with different numerical apertures between the input and output fibers [21], or in launching conditions [22], among others, has been widely discussed. A geometrical model is used to illustrate the induced loss of dust contamination and radial offset misalignment between multimode fibers in intra-vehicle networks [23] and combining lateral and angular misalignments in [24]. The coupling from free space to multimode fibers and misalignment mitigation techniques in spatial division mode multiplexing links based on orbital angular momentum modes are also under discussion, but for links of kms [25,26]. A model describing the influence of tilting misalignment between a radiating object and the optical fiber gathering the radiated power is analyzed in [15]. None of them provides the analysis of a 3D-orientation misalignment between a radiating object and an optical fiber, specifically the directional dependence of lateral displacement under tilted conditions. This is particularly relevant for light collection in off-axis pyrometers.

In this work, we present for the first time a model describing the influence of the OF-optic pyrometer direction of displacement under tilting conditions in finite hot surface temperature measurements requiring high spatial resolution. A simultaneous three-axis misalignment is considered. From this model, it is possible to define those angles and distances that reduce measurement errors due to misalignments for different hot object sizes and types of optical fibers.

## 2. Theoretical Basis and Model Development

This section details the mathematical-geometrical model of an optical fiber pyrometer, which differs from previous approaches [13,15] in the integration of the misalignment between the fiber-optic axis and the center of the emitting surface or target.

The mathematical procedure for calculating the power collected by the optical fiber from each differential surface element (dS_T_) remains consistent with the previous model [15]. However, several modifications are introduced to incorporate these new capabilities, as detailed below. The model is valid for both scenarios, where the tilting angle (θ), the angle between the normal of the emitting target surface and the fiber axis, is zero, and for scenarios with a non-zero tilting angle.

We present two previous models [13,15], in which the emitting surface is a circle with its center aligned with the fiber-optic axis. The first model [13] assumes that the normal of the emitting surface is aligned with the fiber-optic axis, corresponding to a zero-tilting angle. Figure 1 shows the schematic of this approach.

This model calculates the temperature measurement, which critically depends on the relationship between the target size and the minimum distance between the fiber and the target (t), as well as the optical fiber parameters, core radius (r_F_), and numerical aperture (NA).

The optical power coupled into the fiber is calculated by integrating the power emitted by each differential element (dS_T_) of the Lambertian surface. The power contribution from each dS_T_ is calculated by integrating over the intersection area between the fiber core and the projection of its NA-defined light cone onto the end-face plane of the fiber. This model categorizes the integration into three sequences, based on the relative sizes of the fiber core radius (r_F_) and the NA-defined projection radius (r_βmax_), both of which are constant for every differential element dS_T_.

A summary of all parameters discussed in Figure 1, as well as a comprehensive list of the parameters with their meanings and symbols discussed in this manuscript, is included next, in Table 1.

A significant drawback of this model is that it neglects the impact of misalignment—represented by a non-zero tilting angle (θ ≠ 0)—between the fiber axis and the normal of the target surface. This misalignment can lead to considerable errors in dynamic or non-ideal scenarios, such as those involving rotating machinery or vibrating systems. To address this limitation, a second model was presented [15], which extended the prior work by integrating the tilting angle (θ) in the calculation. Figure 2 depicts the schematic of this second model. This case includes the analysis of considering an off-axis pyrometer placed at a tilting angle θ and a non-tilting surface, as in [10,11].

As in the first model, the power coupled into the fiber is calculated by integrating the power emitted by each element dS_T_. Power contribution by each dS_T_ is calculated by integrating the intersection between the fiber core and the light cone projected by the core on the fiber plane (circle with r′_βmax_ in Figure 2). The inclusion of the angle θ modifies this model in some respects, and not only in the spectral power coupled equation. Here, we do not have integration sequences such as the previous one because the light cone projected circle with radius labeled now as r′_βmax_, is not constant for each dS_T_. This changes the relative sizes between r_F_ and r′_βmax_ for each dS_T_, avoiding the integration sequences. Now we have integration cases for each dS_T_.

The distance (r′) between the centers of the fiber-optic core and the circle of the NA-defined projection cone over the fiber end cutting plane shall be recalculated for each dS_T_. Once r′_βmax_ and r′ are calculated, the integration case, with its radial (u_min_, u_max_) and azimuthal (δ_min_, δ_max_) integration limits, is well-defined so that the contribution power of this dS_T_ can be found. Then, the procedure is repeated for the next dS_T_, adding this new contribution to the previous one.

The new model adds new functionalities, including misalignment between the fiber-optic axis and the center of the emitting target surface, while also introducing the surface tilting feature (θ ≠ 0). Figure 3 shows the schematic of the new model illustrating the target misalignment.

The optical power collected by the fiber is determined using the same procedure as in the previous models, by integrating the power contributed by each differential element dS_T_ of the target surface. The optical power supplied by any given element is calculated from the intersection between the fiber core and the light cone projected by each dS_T_ over the fiber end cut plane. The integration cases remain the same as in the previous case, depending on the distance between the center of the fiber core and the center of the NA-defined projected light cone (labeled r″ in Figure 3).

The misalignment is introduced as a horizontal displacement (ΔS) of the center of the target surface from its aligned position, occurring in a direction defined by the angle α. Both ΔS and α are measured in the x′–y′ plane, which is coincident with the end-face plane of the fiber and parallel to the target surface when θ = 0. Figure 4 shows a top-down view of this lateral shift, including the black circle (representing the fiber core with radius r_F_) and the red circle (representing the projected light cone with radius r′_βmax_ from a dS_T_). The dashed blue line represents the tilted emitting target, which is shown as an ellipse due to its inclination angle (θ) relative to the x-axis.

This displacement changes two key parameters (see Figure 4), the distance between the centers of the fiber-optic core and the projected light cone. Which changes from its original value r′ to the new value and label r; and the angle that this distance vector forms with the x-axis on the fiber end plane, now denoted as φ″ (see Figure 4).

The value r″ and φ″ are now given by:(1)$$r^{\prime \prime }\;=\;{\left({r^{\prime }}^{2}+\Delta S^{2}+2r^{\prime }\Delta S\;\cos\left(\alpha -\varphi \prime \right)\right)}^{1/2}$$(2)$$\varphi^{\prime \prime }=\mathrm{Arg}\left(\;\left(r^{\prime }\;\cos\left(\varphi ’\right)+\Delta S\;\cos\left(\alpha \right)\right)+j\;\left(r^{\prime }\;\sin\left(\varphi ’\right)+\Delta S\;\sin\left(\alpha \right)\right)\right)$$
where ΔS is the magnitude of the displacement, α is the angle of this displacement, r′ is the distance between the center of the target object and the center of the circle illuminated by the projected light cone from dS_T_ (both measured on the x′–y′ plane), φ′ is the angle between the vector r′ and the x-axis and j is the imaginary unit. The Arg() function returns the phase of the complex number enclosed within the parentheses. This function resolves the ambiguity of the standard arctangent function, tan^−1^(), for angle values from π/2 to 3π/2. In (2), real and imaginary parts are the components of the vector r″ along the x′ and y′ axes.

The value of r′ and φ′ are given by:(3)$$r^{\prime }\;={\left({\left(r\;\cos\left(\varphi \right)\cos\left(\theta \right)\right)}^{2}+{\left(r\;\sin\left(\varphi \right)\right)}^{2}\right)}^{1/2}$$(4)$$\varphi^{\prime }=\mathrm{Arg}\left(r\;\cos\left(\varphi \right)\cos\left(\theta \right)+j\;r\;\sin\left(\varphi \right)\right)$$
where r and φ are the radial and azimuthal coordinates of dS_T_ in the plane of the target (x–y plane) and θ is the tilting angle of the target. In (4), the real and imaginary parts are the r′ components over the x′ and y′ axes.

The light coupled to the fiber by each dS_T_ is calculated by integrating the spectral radiance of the Lambertian emitting surface differential over the intersection area of the black circle (fiber core with r_F_ radius) with the red circle (light rays with acceptance angles coupling to a fiber are within r′_βmax_ radius), see Figure 4. Depending on relative sizes, these two radii (r′_βmax_, r_F_), and the distance between their centers r″, different scenarios are proposed (as explained in [13], resulting in the integration of arcs, circles, and arcs and circles. The integration limits (u_min_, u_max_) used for the radial coordinates, u, are taken from Table 2 of [15]; these limits are summarized in Table 2. The only difference is the replacement of r′ by r″ due to the introduction of the displacement of the object, calculated using (1).

The new method uses the same formula as in [15] for the integration limits (δ_min_, δ_max_) of the azimuthal coordinates (δ), but with r′ and φ′ replaced by r″ and φ″, and calculated as in (1) and (2). Thus, both expressions are given by:(5)$$\delta_{\min}=\varphi^{\prime \prime }+\pi -\delta_{i}$$(6)$$\delta_{\max}=\varphi^{\prime \prime }+\pi +\delta_{i}$$
where δ_i_ is given by:(7)$$\delta_{i}=\frac{1}{2}\;\left(\delta_{\max}-\delta_{\min}\right)=\cos^{-1}\left(\frac{{r^{\prime \prime }}^{2}+u^{2}-r_{F}^{2}}{2r^{\prime \prime }u}\right)$$

Once the integration case and its limits are obtained, the spectral power coupled to the fiber for each dS_T_ is obtained by calculating the following double integral:(8)$$P_{d\lambda ,{\mathrm{dS}}_{T}}=\overset{u_{\max}}{\underset{u=u_{\min}}{\int }}\overset{\delta_{\max}}{\underset{\delta =\delta_{\min}}{\int }}L\left(\lambda ,T\right)\frac{\left[u\;\cos\left(\delta \right)\sin\left(\theta \right)\;+t^{\prime }\cos\left(\theta \right)\right]t^{\prime }u}{{\left({t^{\prime }}^{2}+{\;u}^{2}\right)}^{2}}d\delta \mathrm{du}$$
where L(λ,T) is the spectral radiance of the Lambertian emitting object, and t′ the shortest distance between the differential element (dS_T_) and the end cut plane of the fiber.

Finally, to consider the contributions of all differential surfaces in the calculus of the optical power, an additional double integration shall be calculated, and given by:(9)$$P_{d\lambda }=\int_{r=0}^{r_{T}}\int_{\varphi =0}^{2\pi }P_{d\lambda ,{\mathrm{dS}}_{T}}\left(r,\varphi ,\theta \right)\mathrm{rd}\varphi \mathrm{dr}$$
being r_T_ the emitting target radius, r and φ the radial and azimuthal coordinates of each dS_T_ over the object, respectively, and P_dλ_ the spectral power from the emitting target coupled to the fiber-optic core.

The spectral radiance L(λ,T), can be represented by the following equation:(10)$$L\left(\lambda ,T\right)=\varepsilon \left(\lambda ,T\right){\;L}_{b}\left(\lambda ,T\right)$$
where L_b_(λ,T) is the spectral radiance of the black body, ε(λ,T) is the emissivity of the object, λ is the wavelength, and T is the object temperature.

In this new model, we can consider the angular dependence of the emissivity for non-polarized radiation, providing a new emissivity value using the mean value of the Fresnel equations [10], for each specific tilting angle.

## 3. Simulation and Discussion

This section presents a complete numerical analysis of the proposed new fiber-optic pyrometer model, focusing on its performance under realistic misalignment conditions. The simulations are designed to validate the robustness of the misalignment model by quantifying power collection efficiency by changing displacement (ΔS) and angular (α) offsets between the fiber axis and target surface, characterizing measurement errors induced by combined tilting (θ) and lateral misalignment effects to simulate real industrial scenarios.

Simulations are performed using an enhanced model code developed in MATLAB R2025b, taking a standard graded index glass optical fiber with a core diameter of 62.5 µm and an NA of 0.275 for a wavelength range from 1430 nm to 1700 nm. Those parameters correspond to an experimental optical fiber pyrometer shown in [4,12]. For all simulations, the target temperature is fixed at 1000 °C. No angular dependence on the emissivity of the target object is considered, using an emissivity value of 1. Any other type OF and temperature can be modeled, although some limitations are discussed in Section 3.2.

### 3.1. Robustness of the Misalignment Model

In this set of simulations, we quantify power collection efficiency under different displacement (ΔS) conditions at specific angular directions (α) for a homogeneous target while maintaining the tilting angle (θ) at 0°.

The first simulation depicts the gathered power vs. displacement for different displacement directions (α): 0°, 45°, 90°, see Figure 5. The emitting object radius (r_T_) is 400 μm, and the minimum distance from the center of the target to the fiber end plane (t) is 150 µm. It demonstrates that the gathered optical power becomes independent of the displacement direction (α), decreasing from a constant power to zero with the same slope regardless of the displacement angle (α).

This invariance occurs because the intersections between the gathering circle created by the illumination of the acceptance cone of the fiber in the plane of the target object (with r_NA_ radius), and the object are equivalent intersections for the same displacement (ΔS) regardless of displacement direction (α). Only the dS_T_ differentials of the emitting object, within the r_NA_ radius circle, are able to couple light to the fiber. This behavior is explained in Figure 6.

It shows that when the gathering circle (r_NA_ radius) is completely in the object circle (r_T_ radius), the power gathered by the fiber is maximum (slightly above 1.82 μW), and begins to decrease when: ΔS = r_T_ − r_NA_ = 326 μm is reached, as can be seen in Figure 5.

Figure 7 shows optical power versus displacement at a distance of 150 μm for two target objects (r_T_): 30 and 60 μm, both lower than the gathering circle at α = 45°.

As seen in the figure, the optical power gathered by the optical fiber is lower for the smaller object, and both curves are monotonically decreasing. The maximum optical power is obtained for ΔS = 0, but in both cases, it is lower than the optical power (1.82 μW) obtained in the simulation shown in Figure 5. This phenomenon is due to the smaller area of the gathering circle that covers the object, which leads to a lower gathered optical power. In both cases, the monotonically decreasing function is due to the Lambertian emission of the surfaces. For larger displacements, the fiber will see the emission surface at increasingly larger angles, receiving a lower intensity emission.

Figure 8 depicts the geometrical description of a displacement (ΔS) for the two objects simulated in Figure 7. No optical power is gathered by the optical fiber when ΔS = r_T_ + r_NA_ is reached, with these values of 134.15 μm and 104.15 μm for the bigger and smaller objects. Note that for values close to these displacements, the power is so small that it appears to be zero, but it is not.

### 3.2. Combined Tilting and Lateral Misalignment Effects

This section examines the non-trivial interaction between lateral displacement (ΔS) and surface tilt (θ), where the directional dependence of displacement (α) becomes critical.

The simulation shown in Figure 9 depicts the optical power gathered by the optical fiber versus displacement at a tilting angle (θ) of 60° for different displacement directions (α): 0°, 45°, 90°. The emitting object radius (r_T_) is 80 μm, and the minimum distance from the center of the target to the fiber end plane (t) is 150 µm.

Figure 9 reveals three key phenomena observed. On one side, there is a different behavior of the gathered optical power for each one of the different displacement directions (α). On the other hand, the slope of optical power decrement versus displacement changes with displacement direction, being greater for α = 0°. This implies that the alignment tolerance and the displacement threshold at which optical power is close to zero are also lower for α = 0°. Finally, there is a local maximum at a specific ΔS for α = 0° and 45°. The first and second phenomena can be explained using the geometric representation of the displacement shown in Figure 10. Due to the tilting, the projection of the circular object onto the plane becomes an ellipse. This causes the intersections of the target with the gathering circle for the same displacement at different directions, which are not the same as those shown in Figure 6; here, the intersection areas are different.

The third phenomenon responds to the Lambertian emission of an object and how an optical fiber sees the normal direction of a tilted target surface, optimally aligning itself while receiving more power emissions. At α = 0 for a fixed tilting angle, the displacement influence is higher than in other displacement directions. Meanwhile, the gathered optical power starts to decrease at around 15 µm (1/10 of the 150 µm minimum distance from the target center to the fiber end plane). It is important to point out that no angular dependence of emissivity is included in this analysis. The tilting angle dependence of the gathered optical power by the fiber-optic pyrometer for dielectric materials will be the same as described above, as their emissivity decreases monotonically with tilting angle [11].

In the case of metals, depending on the extinction coefficient, increasing values of emissivity can be seen for increasing tilting angles [10] that could modify previous conclusions for high tilting angles.

Finally, we performed simulations of the optical power gathered by the fiber-optic pyrometer for tilting angles of 30° and 40°, where emissivity can be considered constant, for different displacements. Results are shown in Figure 11. The same behavior previously reported is shown, but the power decrement starts at greater displacements, in this case around 1/6 of the minimum distance from the center of the target to the fiber end plane of 150 µm and an object radius of 80 µm. The model allows recalibration of the off-axis fiber-optic pyrometer depending on the tilting angle required to avoid blocked or impractical areas while considering lateral misalignments.

### 3.3. Temperature Error Analysis and Model Limitations

This section discusses the relation between misalignments and temperature errors, along with the limitations of the model.

To analyze the temperature errors related to deviations in gathered optical power due to misalignments, a new simulation is performed using parameters based on typical industrial application values. It is considered a tilting angle of 30°, the worst-case direction (α = 0), and a multimode silica fiber with a numerical aperture of 0.275 located at 150 µm. The model also takes into account the size of the hot object whose temperature is to be measured. Different powders used in advanced manufacturing have different particle sizes, measured as their spherical diameter, that typically range from 20 mm to 200 mm [7,27]. In this simulation, we consider an object diameter of 160 mm. To check the temperature error that can be achieved for different displacement misalignments, we consider an initial temperature of 1000 °C. The simulation results are shown in Figure 12.

It is also simulated the reference experiment, considering the same fiber-optic pyrometer perfectly aligned (α = θ = 0) at a distance of 150 μm from the hot object with a radius of 80 μm. The optical power gathered by the fiber-optic pyrometer at different temperatures is shown in Table 3. The same gathered optical powers are marked on Figure 12 for different displacements, considering a temperature of 1000 °C.

According to Figure 12, there is a gathered optical power of 1.718 μW for a temperature of 1000 °C, at a displacement of 30 μm with a tilting angle of 30°. Meanwhile, in the aligned pyrometer, optical power is gathered for a temperature of 990 °C (see Table 3). Then errors in temperature below 1% or 10 °C require displacements below 30 μm. The error increases up to 5% for a 40 μm displacement.

In general terms for a tilting angle q, the ellipse semi-minus axis of the object projection on the optical fiber plane equals r_T_cosθ and the critical lateral displacement where light is not fully collected by the circle defined by the numerical aperture radius equals r_T_cosθ − r_NA_, see Figure 10, in the case under study, r_NA_ = 74 μm and r_T_cosθ = 69 μm meaning a displacement of around 5 μm. On the other hand, no light is detected if the displacement equals r_T_cosθ + r_NA_ = 143 μm. The previous results are in accordance with this analysis.

On the other hand, the limitations of the model on the temperature range depend mostly on the different parameters of the application as a whole. More specifically, if the optical fiber is very close to the hot surface, the maximum temperature that the fiber can withstand has to be considered. On the other hand, after the light propagation through the optical fiber, the signal must be detected, so the type of photo detector used must be considered. Typical values for pyrometers using multimode silica fibers and InGaAs photodiodes range from 200 to 1200 °C [12], while using specialty optical fibers, such as sapphire fibers, the range could be extended up to 2000 °C [28]. On the other hand, any type of optical fibers can be used as long as you know the numerical aperture, radius, and with a small attenuation in the range of tenths of meters. We have mostly made tests with silica single-mode and multimode fibers with numerical apertures ranging from 0.14 to 0.3 and radius from 8 to 400 µm. Finally, an object with a Lambertian radiation pattern is considered.

## 4. Conclusions

For processes requiring accurate control and quality assurance, such as steel annealing additive, manufacturing, or advanced machinery, lateral and angular misalignments can introduce variability and instability into the temperature measurements. Models describing the effects of those misalignments are critical for recalibration purposes or proper alignment, as in off-axis pyrometers in additive manufacturing, where on-axis viewing is blocked or impractical. As a solution to that problem, this paper shows a model describing the influence of the optical fiber pyrometer direction of displacement under tilted conditions in finite hot surface temperature measurements requiring high spatial resolution. The model can simulate simultaneous three-axis misalignments and allows the integration of emissivity. Due to the tilting of a circular object, the projection of it on a plane is an ellipse that breaks the symmetry in the direction of displacement. Results show that errors in temperature below 1% or 10 °C require displacements below 30 μm for a 160 µm object size and a 30° tilting angle. The error increases up to 5% for a 40 μm displacement. For a fixed tilting angle, the influence of displacement is greater in specific directions. Meanwhile, the tilting angle influence increases in higher tilting angles for the same direction of displacement. The model can be used for different types of optical fibers with different core diameters and numerical apertures, considering the effect of the directional dependence of lateral displacement under tilted conditions for different target object sizes, distance from the tip of the fiber to the measurement surface, and a wide temperature range. The model has been applied to analyze single-color optical-fiber pyrometers, which are often more robust to the influence of noise, but it can also be extended to multiple spectral bands.

## Figures and Tables

**Figure 1 sensors-25-07011-f001:**
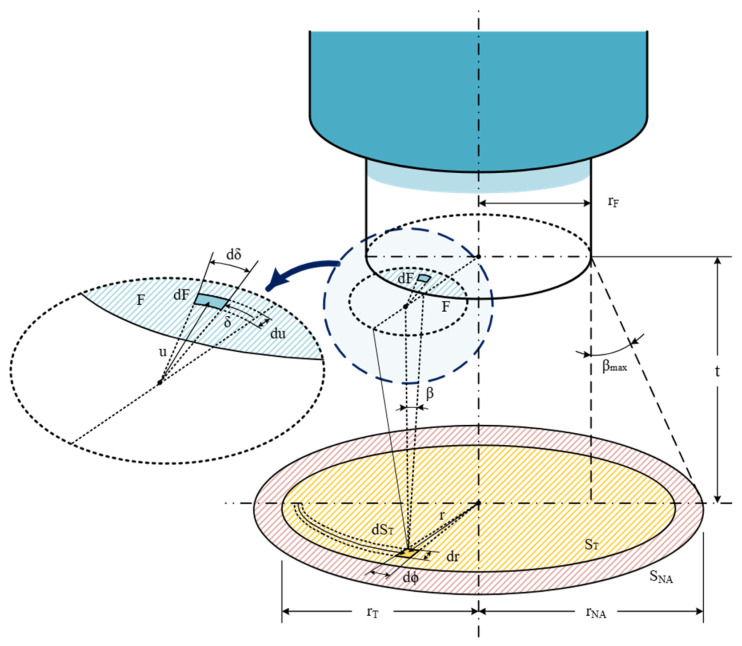
Schematic of the first model fiber-optic pyrometer aligned with the target surface, showing the model variables [15]. Showing surface of OF numerical aperture projection (red striped), surface of the target (yellow striped) and differential target surface projection on the plane of OF surface (blue striped).

**Figure 2 sensors-25-07011-f002:**
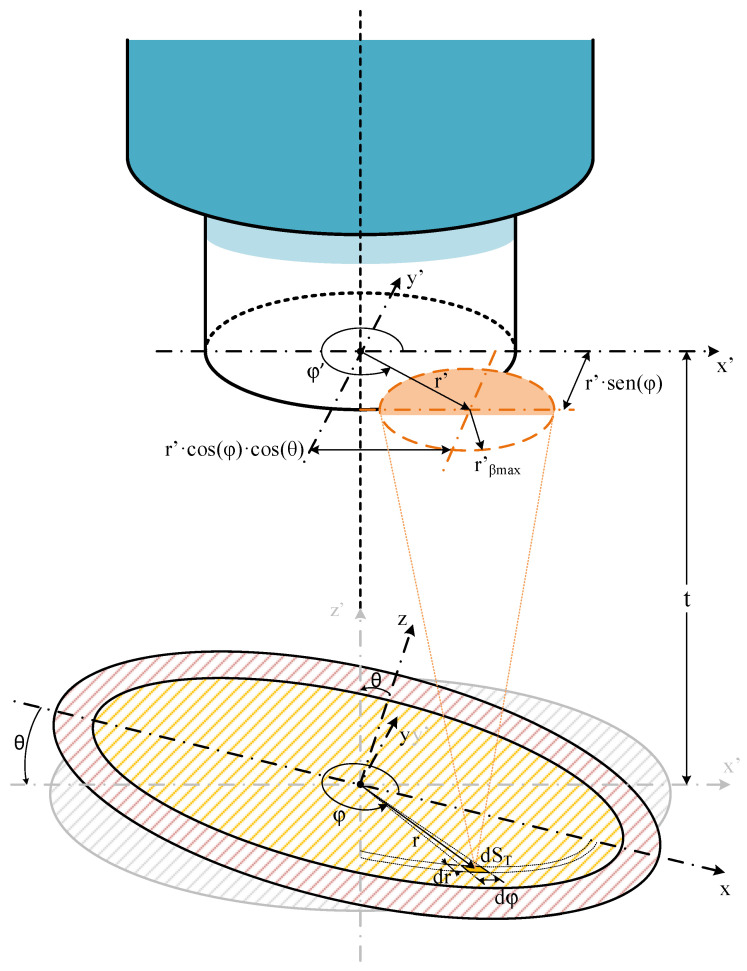
Schematic of the second model fiber-optic pyrometer aligned with the target surface, with a tilting angle (θ), showing the model variables [15]. Showing surface of OF numerical aperture projection (red striped), surface of the target (yellow striped) and differential target surface projection on the plane of OF surface (orange striped).

**Figure 3 sensors-25-07011-f003:**
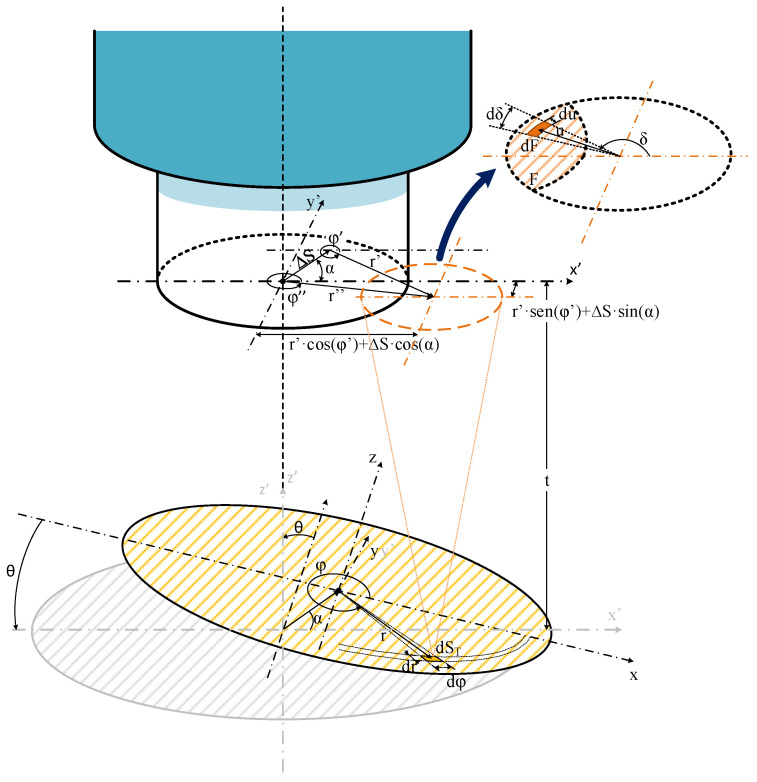
Schematic of the new model fiber-optic pyrometer, with a tilting angle (θ), applying a displacement (ΔS) at an angular direction (α), showing the model variables. Showing surface of the target (yellow striped) and differential target surface projection on the plane of OF surface (orange striped).

**Figure 4 sensors-25-07011-f004:**
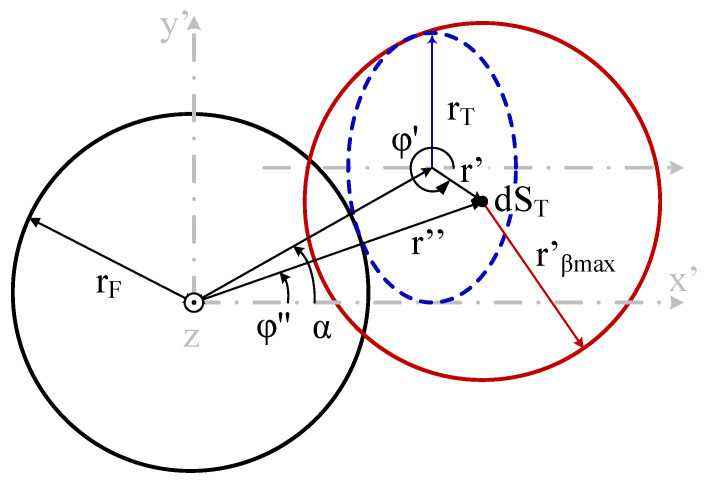
Schematic of the fiber-optic core (black) and the projection of the NA-defined cone on the fiber end plane (red), together with the displacement (ΔS) at a direction angle (α), of the object (dashed blue).

**Figure 5 sensors-25-07011-f005:**
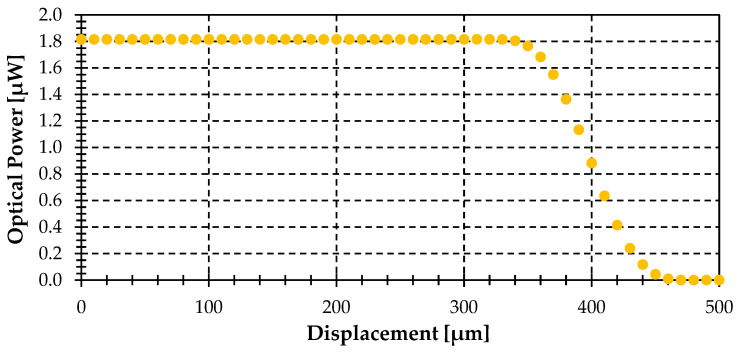
Optical power gathered by the fiber-optic pyrometer versus displacement (ΔS) for different displacement directions (α): 0° (black), 45° (green) and 90° (red), assuming an object radius (r_T_) of 400 μm, a distance (t) of 150 µm, a tilting angle (θ) of 0° and finally, a temperature of 1000 °C. NOTE: The three plots are overlapped.

**Figure 6 sensors-25-07011-f006:**
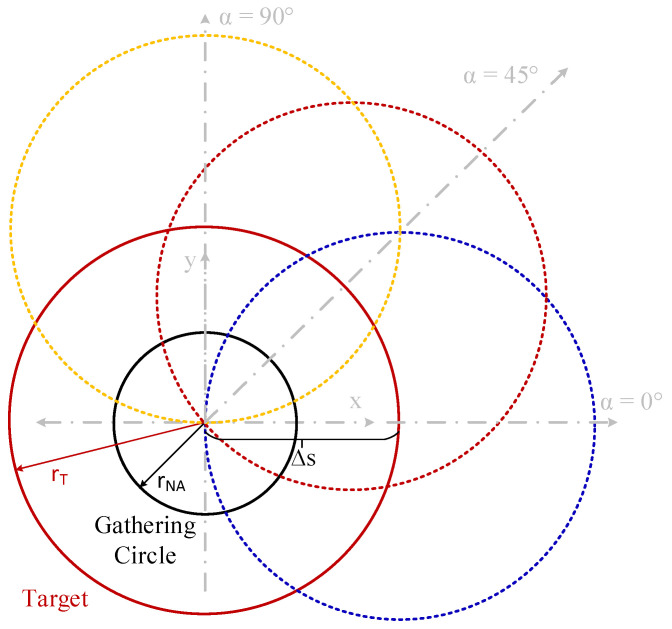
Geometrical description of the displacement (ΔS) at different directions (α): 0° (dashed blue), 45° (dashed red) and 90° (dashed yellow) for the case shown in Figure 5, showing the gathering circle with r_NA_ radius of 74 μm (black) and target object with r_T_ radius of 400 μm (red) (not at scale).

**Figure 7 sensors-25-07011-f007:**
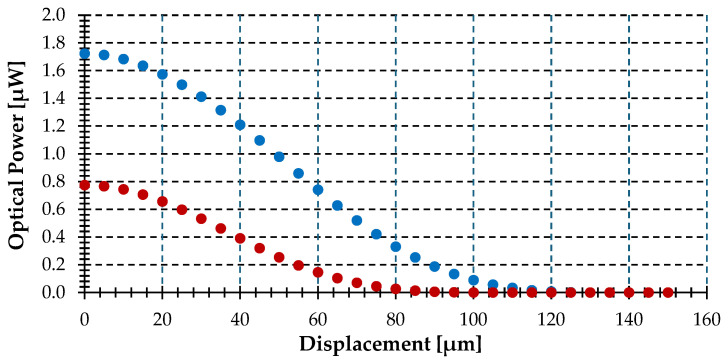
Optical power gathered by the fiber-optic pyrometer versus displacement (ΔS) for different object radius (r_T_): 30 μm (red) and 60 μm (blue), assuming an object radius (r_T_) of 60 μm, a distance (t) of 150 µm, a tilting angle (θ) of 0°, a displacement direction (α) of 45° and a temperature of 1000 °C.

**Figure 8 sensors-25-07011-f008:**
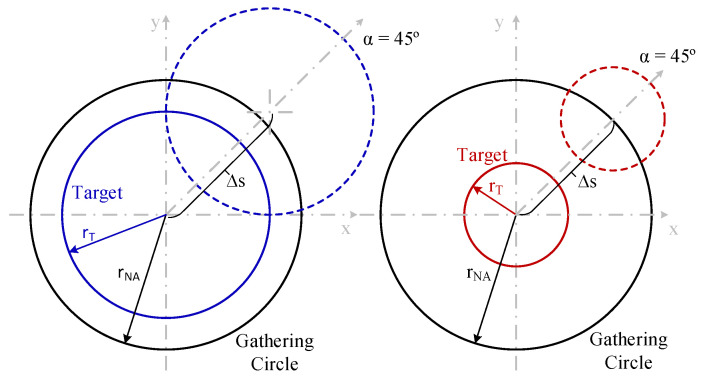
Geometrical description of a displacement (ΔS) at direction (α) of 45° for the two objects simulated in Figure 7, showing the gathering circle with r_NA_ radius of 74.15 μm (black) and target objects with r_T_ radii of 60 μm (blue) and 30 μm (red) (not at scale).

**Figure 9 sensors-25-07011-f009:**
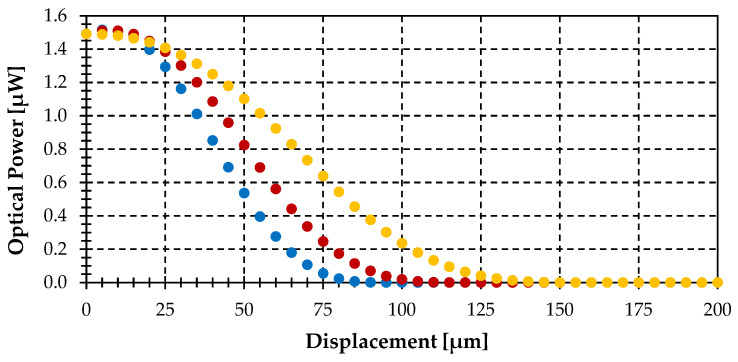
Optical power gathered by the fiber-optic pyrometer versus displacement (ΔS) for different displacement directions (α): 0° (blue), 45° (orange) and 90° (red), assuming an object radius (r_T_) of 80 μm, a distance (t) of 150 µm, a tilting angle (θ) of 60° and finally, a temperature of 1000 °C.

**Figure 10 sensors-25-07011-f010:**
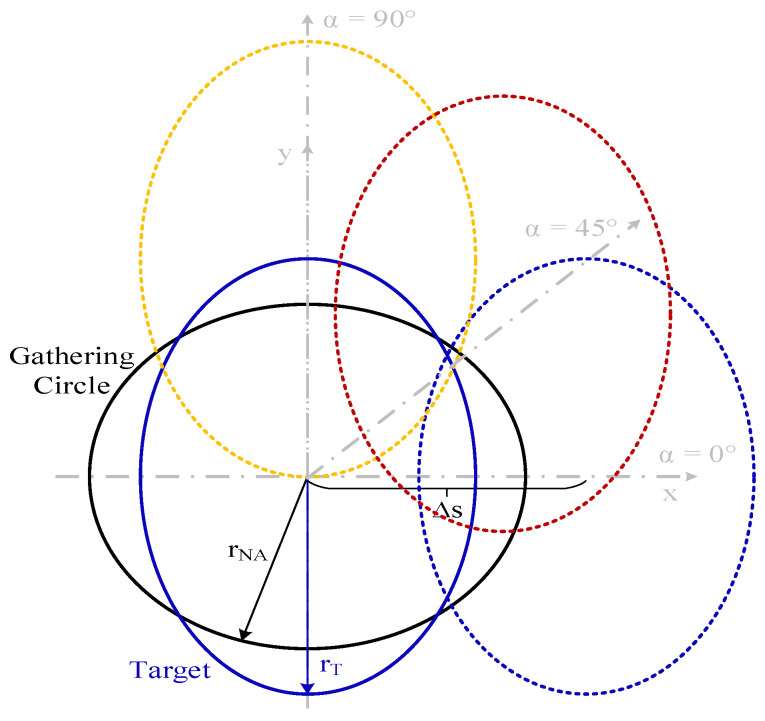
Geometrical description of a displacement (ΔS) at different directions (α): 0° (dashed blue), 45° (dashed red) and 90° (dashed yellow), with a tilting angle θ of 60° for the simulations shown in Figure 9, showing the gathering circle with r_NA_ radius of 74.15 μm (black), and target object with r_T_ radius of 80 μm (blue) (not at scale).

**Figure 11 sensors-25-07011-f011:**
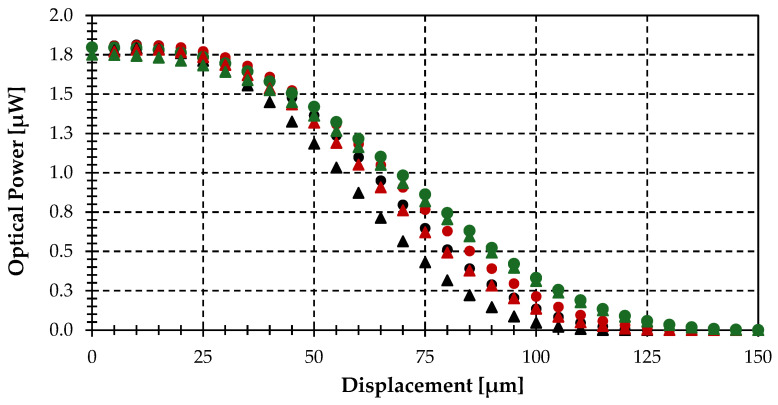
Optical power gathered by the fiber-optic pyrometer vs. displacement (ΔS) for different directions (α): 0° (black), 45° (green) and 90° (red), and tilting angles (θ) of 40° (circle), 30° (triangles), assuming an object radius (r_T_) of 80 μm, a distance (t) of 150 µm and finally, a temperature of 1000 °C.

**Figure 12 sensors-25-07011-f012:**
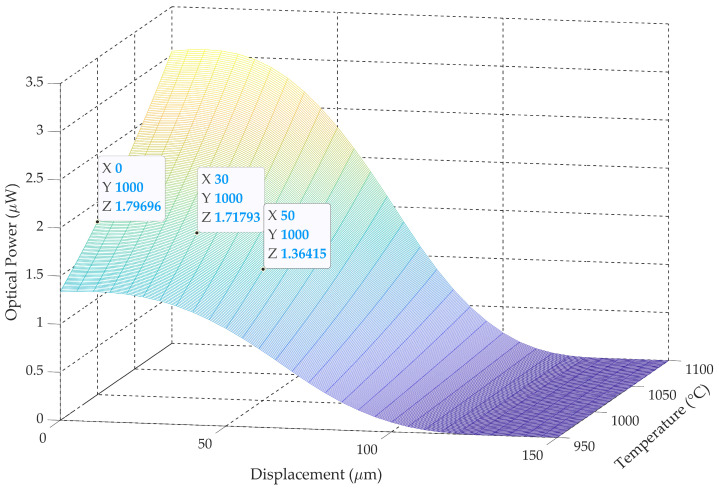
Optical power gathered by the fiber-optic pyrometer vs. displacement (ΔS) and temperature (T), assuming a direction (α) of 0°, a tilting angle (θ) of 30°, an object radius (r_T_) of 80 μm, a distance (t) of 150 µm. Color degradation from yellow (highest power) to blue (lowest power).

**Table 1 sensors-25-07011-t001:** Explanation of parameters of models and their meanings updated from [15].

Variable	Meaning	Variable	Meaning
dS_T_	Differential element of the target surface	r_T_	Radius of the target
r_F_	Optical fiber (OF) core radius	β	Angle between the normal to dS_T_ and the vector from dS_T_ to each solid angle differential in the intersection of the circles with radii r_F_ and r_βmax_ or r′_βmax_
r_βmax_, r′_βmax_ *	Radius of the circle defined by the light cone projection, due to OF NA	dF	Differential element of area of circle with radius r_βmax_ or r′_βmax_
t, t′ *	Minimum distance from dS_T_ to fiber end plane on each model	u	Radial coordinate of each differential element of area dF
r	Radial coordinate of dS_T_ on the plane of the target	δ	Azimuthal coordinate of each differential element of area dF
r′	Distance of the projected object center on the fiber end plane, to the center of a circle with radius r_βmax_ or r′_βmax_	θ	Angle between the fiber axis and the normal to the emitting target surface
r″	Distance between the centers of the circles with radii r_F_ and r′_βmax_	ΔS	Displacement of the center of the target object, in a parallel plane to the fiber end plane (x′–y′)
φ	Azimuthal coordinate of dS_T_ on the target plane on each model	α	Angle of the direction of displacement of vector ΔS
φ′	Angle between r′ and x-axis on the fiber end plane	r_NA_	Radius of the gathering circle defined by the OF field of view due to NA, on the target plane
φ″	Angle between r″ and x-axis on the fiber end plane	β_max_	Maximum acceptance angle OF

* The variables t and r_βmax_ are used in the first model, and t′ and r′_βmax_ are used in the second and third models. They are defined in the same way, but they are calculated differently.

**Table 2 sensors-25-07011-t002:** Integration limits of radial coordinates u_min_, u_max_.

		0 < r″ < r_F_ − r′_βmax_		r_F_ − r′_βmax_ < r″ < r_F_		r_F_ < r″ < r_F_ + r′_βmax_
r′_βmax_ < r_F_	Int. Sit.	Circum.		Circum.	Arcs	Arcs
	u_min_	0		0	r_F_ − r″	r″ − r_F_
	u_max_	r′_βmax_		r_F_ − r″	r′_βmax_	r′_βmax_
r_F_ < r′_βmax_ < 2r_F_	Int. Sit.	Circum.	Arcs	Circum.	Arcs	Arcs
	u_min_	0	r_F_ − r″	0	r_F_ − r″	r″ − r_F_
	u_max_	r_F_ − r″	r_F_ + r″	r_F_ − r″	r′_βmax_	r′_βmax_
2r_F_ < r′_βmax_	Int. Sit.	Circum.	Arcs	Arcs		Arcs
	u_min_	0	r_F_ − r″	r″ − r_F_		r″ − r_F_
	u_max_	r_F_ − r″	r_F_ + r″	r″ + r_F_		r′_βmax_

**Table 3 sensors-25-07011-t003:** Optical power gathered by aligned fiber-optic pyrometer (α = θ = 0), r_T_ = 80 μm.

Temperature (°C)	998	990	951
Optical power (μW)	1.796	1.718	1.364

## Data Availability

The original contributions presented in this study are included in the article. Further inquiries can be directed to the corresponding author.

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
