# Peer review of "Misalignment Effects on Power Gathered by Optical Fiber Pyrometer"

_sensors, 2025, doi:10.3390/s25227011_

Round 1

Reviewer 1 Report

Comments and Suggestions for Authors

This article presents a model of a fiber-optic pyrometer that accounts for spatial misalignments. The relevance of this work is related to the requirements of precise temperature measurements in additive manufacturing. The model demonstrates an approach to simulation for triaxial displacements. This work addresses the important issue of calculating for misalignment in fiber-optic pyrometers but requires further development.

Point 1. The abstract states that lateral misalignments influence is negligible for values ​​below 1/10 of the minimum distance from the center of the target to the fiber end plane. A specific distance value with error (RMS/MSE, etc.) in SI units is required.

Point 2. The introduction allows the reader to form a general idea of ​​the current state of the art. However, this section contains 7 refs to the authors of this work out of the 19 listed in the bibliography. The introduction could be improved by diversifying the research groups involved in optical radiation misalignment research, for example, https://doi.org/10.1038/s42005-025-02323-7; https://doi.org/10.1364/JOT.90.000236.

Point 3. Unify wording and notation:

A) Figures 1 and 2 are poorly presented. The parameter notations (angles, distances), especially in Figure 1, should be made more readable, and they should be clearly indicated in the text of the article.

B) For the new model from Section 2.2, explanations for most of the parameters indicated in Figure 3 are also missing.

C) Str.159-160 "with the same formulas, however, some parameters within these formulas must be re-evaluated to consider the displacement factor." It is necessary to clearly indicate which eq. the authors are referring to.

D) The text of the article does not specify the value or range of the radiation wavelength and other model parameters.

Point 4. It is necessary to quantify the limitations of the model: determine temperature ranges (e.g., 500-2000°C) and fiber types.

Point 5. It is recommended to formalize the criteria for acceptable misalignments: establish maximum displacement values ​​for typical industrial applications (e.g., ±0.5 mm at angles up to 30°).

The model is of practical value for industrial pyrometry. After revision, the article may be recommended for publication. Consideration of the combined effects of misalignments is the most significant result.

Major revision

Author Response

This article presents a model of a fiber-optic pyrometer that accounts for spatial misalignments. The relevance of this work is related to the requirements of precise temperature measurements in additive manufacturing. The model demonstrates an approach to simulation for triaxial displacements. This work addresses the important issue of calculating for misalignment in fiber-optic pyrometers but requires further development.

Comment 1. The abstract states that lateral misalignments influence is negligible for values ​​below 1/10 of the minimum distance from the center of the target to the fiber end plane. A specific distance value with error (RMS/MSE, etc.) in SI units is required.

RESPONSE:

Thank you for your comment. The value provided in the abstract is considering a tilting angle of -60º and a distance from the center of the hot object (target) to the fiber end plane of 0.15 mm, meaning a negligible influence of lateral misalignments of 15 um, there is no error in this case as it is the result of the simulation for specific data. If we consider a 1 um deviation of that value, the influence of the lateral misalignment is still negligible. The provided data depends on the hot object size and the optical fiber parameters, in this case it is considered an object with a radius of 80 um, or object size of 160 um, and an optical fiber providing a numerical aperture radius at that distance of 74 um. For every case we can have different values, we included this as an example.

We have now modified the abstract according to the reviewer comment and we have included all parameter for the specified case for clarity.

Comment 2. The introduction allows the reader to form a general idea of ​​the current state of the art. However, this section contains 7 refs to the authors of this work out of the 19 listed in the bibliography. The introduction could be improved by diversifying the research groups involved in optical radiation misalignment research, for example, https://doi.org/10.1038/s42005-025-02323-7; https://doi.org/10.1364/JOT.90.000236.

RESPONSE:

Thank you for your comment. We have revised the references provided by the reviewer about optical radiation misalignments effect on some photonics systems. They are related to specific optical beams including the alignment errors influence on vortex beams formation [A] without any optical fiber involved, and the problem of coupling cylindrical vector beams from free space to multimode fibers as part of a spatial division mode multiplexing link of 5 km [B]. In this last case, coupling improvement by using specific meta-devices designs is discussed. In our study, we have used a geometrical model to describe how much light is coupled to an optical fiber considering its numerical aperture, so there is no clear connection with the first work discussed in [A]. Perhaps in the second case, being considered the light coupling from free space to a multimode fiber it can be somehow some relation although we have not estimated how many modes are excited and how they propagate in the multimode optical fiber. The typical optical fiber length considered in our study, although not specifically included in the analysis, of optical fiber pyrometers is not very long of around several meters at most, not of kms as in [B] so attenuation and other aspects of optical fiber propagation are considered negligible. We have now extended our analysis in the state of the art to include some optical coupling analysis including misalignment effects such as [B-H] .

The new text in the introduction is as follows:

The effect of different misalignments on light coupling between optical fibers, as angular misalignment in [C], between dissimilar fibers with tapers [D] or in launching conditions [E] among others have been widely discussed. A geometrical model is used to illustrate the induced loss by dust contamination and radial offset misalignment between multimode fibers in intra-vehicle networks [F], and  combining lateral and angular misalignments in [G]. The coupling from free space to multimode fibers and misalignment mitigation techniques in spatial division mode multiplexing links based on orbital angular momentum modes are also under discussion bur for links of kms [B, H]. A model describing the influence of tilting misalignment between a radiating object and the optical fiber gathering the radiated power is analyzed in [I]. None of them provide the analysis of a 3D-orientation misalignment between a radiating object and an optical fiber, specifically the directional dependence of lateral displacement under tilted conditions which is relevant when collecting light as in off-axis pyrometers.

[A] https://doi.org/10.1364/JOT.90.000236.

[B] https://doi.org/10.1038/s42005-025-02323-7

[C] Opielka, D.; Rittich, D. Transmission loss caused by an angular misalignment between two multimode fibers with arbitrary profile exponents. Appl. Opt. 1983, 22, 991–994.

[D] doi: 10.3390/photonics10050513

[E] doi: 10.1109/JLT.2021.3051458.

[F] https://doi.org/10.1016/j.optcom.2024.130575

[G] https://doi.org/10.3390/s19224968

[H] doi: 10.1109/JLT.2025.3575727

[I] https://doi.org/10.3390/s23198119

Comment 3. Unify wording and notation:

  1. Figures 1 and 2 are poorly presented. The parameter notations (angles, distances), especially in Figure 1, should be made more readable, and they should be clearly indicated in the text of the article.

RESPONSE:

Thank you for your comment. The Figures have been modified according to the reviewer comments. The blurred Figure 1 has been replaced with a high-resolution picture as well as some improvements in the legibility of the information shown in both figures; name of certain parameters, included relevant information for better understanding of the spatial meaning of the figure, among others.

In addition, to better understand all parameters shown in Figure 1, 2 and 3, as well as all formulas and schematics in the manuscript, a comprehensive list of the parameters with their meanings and symbols has been included in new Table I.

  1. For the new model from Section 2.2, explanations for most of the parameters indicated in Figure 3 are also missing.

RESPONSE

Thank you for your comment. The parameters of the new model included in Fig. 3 are described in new Table 1.  The list of the parameters includes their meanings.  This list covers the parameters detailed in the main text and employed in Figures 1, 2, and 3.

  1. 159-160 "with the same formulas, however, some parameters within these formulas must be re-evaluated to consider the displacement factor." It is necessary to clearly indicate which eq. the authors are referring to.

RESPONSE:

Thank you for your comment. The sentence has been removed, as it was not necessary. These formulas are described later in the article in equations (5), (6), and (7).

  1. The text of the article does not specify the value or range of the radiation wavelength and other model parameters.

RESPONSE:

Thank you for your comment. The range of radiation wavelength is from 1,430 to 1,700 nm; these values are included in second paragraph of Section 3: Simulation and Discussion. They are considered as typical wavelength ranges when using some filters in the optical fiber pyrometer, but the model allows to use other values if required. The optical fiber considered is a OM1 silica multimode fiber but again other fibers can be used, we have discussed now the limitations of the model describing the optical fiber types that can be used. About the range of the other parameters, it depends on the application. The emissivity is considered to be equal to 1, but again it can be change. Simulations are performed considering different object sizes, tilting angles, direction angle and displacement.

Comment 4. It is necessary to quantify the limitations of the model: determine temperature ranges (e.g., 500-2000°C) and fiber types.

RESPONSE:

Thank you for your comment. The limitations of the model on temperature range depends mostly on the different parameters of the application as a whole. More specifically, if the optical fiber is very close to the hot surface, the maximum temperature that the fiber can stand has to be considered. On the other hand, after the light propagation through the optical fiber the signal must be detected so the type of photodetector used have to be taken into account. Typical values for pyrometers using multimode silica fibers and InGaAs photodiodes range from 200-1200 º [A] while using specialty optical fibers such as sapphire fibers that range could be extended up to 2000 ºC [B]. On the other hand, any type of optical fibers can be used as far as you know the numerical aperture, radius and with small attenuation in the range of tenths of meters. We have mostly made tests with silica single mode and multimode fibers with numerical apertures ranging from 0.14 to 0.3 and radius from 8 to 400 um. The influence of attenuation and other optical fiber parameters on optical fiber pyrometers is provided in [C, D].

We have included this discussion of the model limitations in new subsection 3.2 Temperature error analysis and model limitations

[A] https://doi.org/10.1109/JSEN.2020.3023240

[B] doi: 10.1109/JLT.2024.3438108

[C]  doi: 10.1109/JSTQE.2016.2627553.

[D] doi: 10.1109/JLT.2015.2513158

Comment 5. It is recommended to formalize the criteria for acceptable misalignments: establish maximum displacement values ​​for typical industrial applications (e.g., ±0.5 mm at angles up to 30°).

RESPONSE:

Thank you for the comment. We have included a new simulation for the case of a tilting angle of 30°, the worst-case direction a=0, and using a multimode silica fiber with a numerical aperture of 0.275 located at 0.15 mm, in order to check the temperature error than can be achieved for different displacement misalignments. The model also takes into account the size of the hot object whose temperature is to be measured. Different powders used in advanced manufacturing have different particles sizes, measured as their spherical diameter, that typically range from 20 um to 200 um [A,B]. In this simulation we consider an object diameter of 160 um.

The results are shown in Fig. 12 and are discussed afterwards.

In general terms for a tilting angle q, the ellipse semi-minus axis of the object projection on the optical fiber plane equals rT cos q  and the critical lateral displacement where light is not fully collected by the circle defined by the numerical aperture radius equals rT cosq-rNA , see Fig. 10, in the case under study, rNA=74 um and rT cosq-=69 um meaning a displacement of around 5 um. On the other hand, no light is detected when the displacement equals rT cosq+rNA =143 um.

[A] DOI: 10.12913/22998624/209176

[B] https://doi.org/10.1016/j.surfcoat.2022.128818

Reviewer 2 Report

Comments and Suggestions for Authors

This manuscript presents a comprehensive geometric-optical model to analyze the effects of combined lateral and angular misalignments on the power collected by an off-axis, single-color optical fiber pyrometer. The work is technically sound, addresses a relevant problem in non-contact thermometry for advanced manufacturing, and is supported by detailed simulations. The model represents a clear and valuable extension of the authors' previous work. I suggest it can be considered for publication after monor revision.

  1. Clearly state the specific gap in the literature—the lack of models analyzing the directional dependence of lateral displacement under tilted conditions—and how this model fills that gap.
  2. The connection between the simulated "gathered power" and the ultimate impact on temperature measurement error is implicit but should be made more explicit. A brief discussion on how a X% change in gathered power translates to a Y K error in temperature (for a given emissivity and wavelength) would significantly strengthen the practical implications of the work.
  3. Improve the quality of Figure 1 (the text in this figure is blurred).
  4. Several figures are referenced before they appear (e.g., Figure 1 is described on Page 3 but is not in the provided excerpt). The captions for Figures 5, 7, 9, and 11 are incomplete without their corresponding plots. The manuscript must be checked to ensure all figures are present and correctly referenced. The axis labels in the described figures (e.g., "Optical Power [W]") should use appropriate units (e.g., µW, as mentioned in the text). : Integrate all missing figures into the manuscript. Ensure all figure captions are self-contained and that all axes are properly labeled with correct units.
  5. Include a discussion (even if qualitative) on how the predicted variations in gathered optical power directly led to errors in temperature reading. This is critical for the application-oriented audience of Sensors.
  6. The conclusion should more forcefully summarize the key novel finding (the critical interaction between tilt and displacement direction) and its practical implication for alignment tolerances in industrial settings.
  7. The manuscript requires thorough proofreading for English grammar and typos (e.g., "monolithically " should likely be "monotonically" on Page 10 ( line 330)).

Author Response

This manuscript presents a comprehensive geometric-optical model to analyze the effects of combined lateral and angular misalignments on the power collected by an off-axis, single-color optical fiber pyrometer. The work is technically sound, addresses a relevant problem in non-contact thermometry for advanced manufacturing, and is supported by detailed simulations. The model represents a clear and valuable extension of the authors' previous work. I suggest it can be considered for publication after minor revision.

Comment 1. Clearly state the specific gap in the literature—the lack of models analyzing the directional dependence of lateral displacement under tilted conditions—and how this model fills that gap.

The introduction has been rewriting clearly showing the gap in literature as recommended by the reviewer.

The new text in the last part of the introduction is as follows:

The effect of different misalignments on light coupling between optical fibers, as angular misalignment in [C], between dissimilar fibers with tapers [D] or in launching conditions [E] among others have been widely discussed. A geometrical model is used to illustrate the induced loss by dust contamination and radial offset misalignment between multimode fibers in intra-vehicle networks [F], and combining lateral and angular misalignments in [G]. The coupling from free space to multimode fibers and misalignment mitigation techniques in spatial division mode multiplexing links based on orbital angular momentum modes are also under discussion bur for links of kms [B, H]. None of them provide the analysis of a 3D-orientation misalignment between a radiating object and an optical fiber, specifically the directional dependence of lateral displacement under tilted conditions which is relevant when collecting light as in off-axis pyrometers.

[A] https://doi.org/10.1038/s42005-025-02323-7

[B] https://doi.org/10.1364/JOT.90.000236.

[C] Opielka, D.; Rittich, D. Transmission loss caused by an angular misalignment between two multimode fibers with arbitrary profile exponents. Appl. Opt. 1983, 22, 991–994.

[D] doi: 10.3390/photonics10050513

[E] doi: 10.1109/JLT.2021.3051458.

[F] https://doi.org/10.1016/j.optcom.2024.130575

[G] https://doi.org/10.3390/s19224968

[H] doi: 10.1109/JLT.2025.3575727

Comment 2. The connection between the simulated "gathered power" and the ultimate impact on temperature measurement error is implicit but should be made more explicit. A brief discussion on how a X% change in gathered power translates to a Y K error in temperature (for a given emissivity and wavelength) would significantly strengthen the practical implications of the work.

RESPONSE:

Thank you for the comment. We have included a new simulation for the case of a tilting angle of 30°, the worst-case direction alpha=0, and using a multimode silica fiber with a numerical aperture of 0.275 located at 0.15 mm, in order to check the temperature error than can be achieved for different displacement misalignments. The model also takes into account the size of the hot object whose temperature is to be measured. Different powders used in advanced manufacturing have different particles sizes, measured as their spherical diameter, that typically range from 20 um to 200 um [A, B]. In this simulation we consider an object diameter of 160 um.

The discussion on the temperature errors related to deviation in gathered power due to misalignments is included in new subsection 2.3. The description of the new simulation with the related parameters and results are shown in Fig. 12 and are discussed afterwards.

In general terms for a tilting angle q, the ellipse semi-minus axis of the object projection on the optical fiber plane equals rT cos q  and the critical lateral displacement where light is not fully collected by the circle defined by the numerical aperture radius equals rT cosq-rNA , see Fig. 10, in the case under study, rNA=74 um and rT cosq-=69 um meaning a displacement of around 5 um. On the other hand, no light is detected when the displacement equals rT cosq+rNA =143 um.

[A] DOI: 10.12913/22998624/209176

[B] https://doi.org/10.1016/j.surfcoat.2022.128818

Comment 3. Improve the quality of Figure 1 (the text in this figure is blurred).

RESPONSE:

Thank you for the comment. The quality of Figure 1 has been improved, replacing the blurred one with a high-resolution picture.

Comment 4.

4.1 Several figures are referenced before they appear (e.g., Figure 1 is described on Page 3 but is not in the excerpt provided).

RESPONSE:

Thank you for the comment. All references and figures have been modified to reference each figure before it appears. A new pdf file is generated.

4.2 The captions for Figures 5, 7, 9, and 11 are incomplete without their corresponding plots.

RESPONSE:

Thank you for the comment. The plots and the figure descriptions have been improved to match the colors with the plots.

4.3 The manuscript must be checked to ensure all figures are present and correctly referenced.

RESPONSE:

Thank you for the comment. As I commented on previous comment, all references and figures have been modified to boost the legibility of the document.

4.4 The axis labels in the described figures (e.g., "Optical Power [W]") should use appropriate units (e.g., µW, as mentioned in the text): Integrate all missing figures into the manuscript. Ensure all figure captions are self-contained and that all axes are properly labeled with correct units.

              RESPONSE:

Thank you for the comment. The following changes have been made in all figures:

  • All figures have been changed to show optical power in µW units as well as their Y-axis label.
  • All figure captions have been checked to be self-contained, including all parameters used to simulate these results.

Comment 5. Include a discussion (even if qualitative) on how the predicted variations in gathered optical power directly led to errors in temperature reading. This is critical for the application-oriented audience of Sensors.

              RESPONSE:

Thank you for the comment. A new subsection 3.2 Error analysis and model limitations is now included where those aspects are discussed.

Comment 6. The conclusion should more forcefully summarize the key novel finding (the critical interaction between tilt and displacement direction) and its practical implication for alignment tolerances in industrial settings.

RESPONSE:

Thank you for the comment and guidance. The conclusions have been rewritten.

Comment 7. The manuscript requires thorough proofreading for English grammar and typos (e.g., "monolithically " should likely be "monotonically" on Page 10 (line 330)).

RESPONSE:          

Thank you for the comment. The referred typo has been corrected and a complete revision of the English grammar and typos has been carried out to improve the legibility of the manuscript.

Round 2

Reviewer 1 Report

Comments and Suggestions for Authors

Accept